# Radiological and Clinical Outcome after Multilevel Anterior Cervical Discectomy and/or Corpectomy and Fixation

**DOI:** 10.3390/jcm7120469

**Published:** 2018-11-23

**Authors:** Paul Oni, Rolf Schultheiß, Kai-Michael Scheufler, Jakob Roberg, Ali Harati

**Affiliations:** Department of Neurosurgery, Klinikum Dortmund, 44145 Dortmund, Germany; paul.oni@klinikumdo.de (P.O.); rolf.schultheiss@klinikumdo.de (R.S.); kai-michael.scheufler@klinikumdo.de (K.-M.S.); jakob.roberg@klinikumdo.de (J.R.)

**Keywords:** anterior cervical decompression and fusion (ACDF), cervical degenerative myelopathy, ossification of the posterior longitudinal ligament (OPLL), lordosis, corpectomy

## Abstract

Background: Multilevel anterior cervical decompression and fixation of four and more levels is a common surgical procedure used for several diseases. Methods: We reviewed the radiological and clinical outcomes after anterior cervical discectomy or corpectomy and fixation of four and more levels in 85 patients (55 men and 30 women) with an average age of 59.6 years. Surgical indication was multilevel cervical degenerative myelopathy and radiculopathy in 72 (85%) patients, multilevel cervical spondylodiscitis in four (5%), complex traumatic cervical fractures in four (5%), metastatic cervical spine tumor in two (2%), and ossification of the posterior longitudinal ligament in three (3%) patients. Results: There were no severe intraoperative complications such as spinal cord or vertebral artery injury or dissection. Seventy-three patients had four, 10 patients had five, and two patients had six anterior cervical level fixations. The visual analog scale (VAS) and Japanese Orthopedic Association (mJOA) scale scores improved (6.9 to 1.3 (*p* < 0.001) and 13.9 to 16.5 (*p* < 0.001), respectively). The Cobb angle increased from 5.7° to 17.6° postoperatively (*p* < 0.001). Secondary posterior fixation was necessary in three cases due to pseudarthrosis. Conclusion: The anterior approach appears to be optimal for ventral compressive pathology and lordosis restoration to the cervical spine. Limitations of multiple level decompression and fixation included increasing pseudoarthrosis rates, especially after corpectomy, and increasing fused level numbers.

## 1. Introduction

The anterior approach for discectomy and fixation described by Smith and Robinson [1] and Cloward [2] have now been routinely used for over five decades. Anterior cervical discectomy and fixation is a standard surgical procedure used for a variety of diseases that affect the anterior and middle cervical columns. The anterior approach allows on-site decompression of the spinal cord and nerve roots in addition to reconstruction of the cervical spine [3,4,5]. In recent years, an increasing number of series have reported good to excellent clinical and radiological outcomes after multilevel anterior cervical spine surgeries involving four or more levels [6,7]. However, there continues to be a shortage of literature on the varying surgical indications and complication rates [8,9]. In this retrospective series, we examined the clinical and radiographic outcomes of 85 cases with multilevel anterior cervical fixation over a 10-year period. To our knowledge, this study represents one of the largest series up to this point.

## 2. Methods

### 2.1. Study Population

This study is a retrospective, descriptive case series of 85 patients (55 men and 30 women) who underwent consecutive four-level or more anterior cervical discectomy and/or corpectomy and fixation. The surgeries were between October 2008 and November 2017 at a single university-affiliated hospital.

Using the hospital’s database, patients’ medical records and radiological images were reviewed. The levels involved ranged from C2 to T2. Clinical outcomes were routinely assessed by the pre- and postoperative modified Japanese Orthopedic Association (mJOA) scores in addition to the visual analog scale (VAS) for neck and arm pain. Radiological outcomes were determined with a plane standing lateral X-ray or reconstructed sagittal computed tomography (CT) scan. Pre- and postoperative cervical lordosis and kyphosis at the last follow-up were measured using the Cobb angle between the lower endplates of C2 and C7. The criteria for determining bone fixation on X-ray and/or CT scan consisted of three parameters: (1) the continued bone trabeculation at the cage/bone interface without lucencies between the cage and bone; (2) no signs of failure of the anterior cervical plate construct; and (3) gross bone formation between the superior and inferior endplates. Radiological evidence of adjacent segment disc degeneration was defined as the presence of disc space narrowing, presence of new or increased osteophyte formation, endplate sclerosis, or increased calcification of the anterior longitudinal ligaments. Four patients without follow-up were excluded. All patients had pre- and postoperative plain cervical X-ray and/or CT scans. All but three patients underwent preoperative cervical magnetic resonance imaging (MRI) for accessing spinal cord compression. Additional details are described in Table 1. The main indication for surgery was multilevel cervical degenerative myelopathy and radiculopathy in 72 patients, two of which had revision due to a failed prior construct. (Figure 1a–c). Most patients had additional single or multilevel kyphotic deformities (Figure 2a–c). Additional indications were multilevel cervical spondylodiscitis with epidural abscess in four, complex traumatic cervical fractures in four, metastatic cervical spinal tumors in two, and ossification of the posterior longitudinal ligament (Figure 3a–d) in three patients. The mean duration of symptoms before surgery was 9.9 months. Seventy-four patients (87%) presented with sensory loss in the upper limb, of which 35 patients reported additional tingling paraesthesia of the upper limbs. Forty-three patients (51%) had loss of motor function involving the upper limb, 18 of which presented with partial tetraparesis. Sphincter dysfunction was evident in 13 patients (15%). A severe gait disturbance was found in 34 patients (40%). Altogether, clinical evidence of spinal cord compression was present in 69 patients (81%) among all patients. The major surgical goals were thorough decompression of the spinal cord and nerve root in addition to restoration of the cervical lordosis.

### 2.2. Surgical Techniques

Under general anesthesia, the patient’s head was placed slightly reclined on a gelatin ring. A 4–6 cm median transversal skin incision was performed. Afterward, the skin with the subcutaneous flap was dissected from the platysma. Vertical splitting of the platysma along its fibers was followed by blunt dissection of the neck soft tissue in a cranial and caudal fashion developing an avascular plane between the esophagus/trachea medially and the sternocleidomastoid muscle/carotid–jugular sheath laterally. All patients underwent surgery via a left-sided approach. In some cases, the omohyoid muscle had to be cut to ease movement of the mobile retractor system and later readjusted at closure. After wide anterior longitudinal exposure of the cervical spine, the longus colli muscles were dissected bilaterally, and a self-retaining mobile retractor was placed in that location. If possible, a reduction of the endotracheal cuff pressure was performed while the retractors were inserted. Preferentially, we prefer starting the decompression at the C4/5 level if possible in order to avoid distraction-related tension and possible postoperative C5 palsy. After placement of Caspar vertebral body distractors, a discectomy was performed. The uncovertebral joint was then carefully resected. Care should always be taken not to injure the vertebral artery. This allows for better segmental distraction and kyphotic deformity repositioning. Using a high-speed drill, the dorsal osteophytes of the cranial and caudal vertebrae were drilled out in a bell-mouth shape. The spinal cord was carefully and thoroughly decompressed with removal of the posterior longitudinal ligament using a Kerrison punch. The decompression was completed by the total removal of the posterior part of the uncovertebral joint, thereby exposing the nerve roots. A corpectomy was performed in case of inability to address dorsal pathology. An intervertebral titanium cage (such as a Cervidisc nach Böker und Schultheiß® (Fa. Weber, Bonn, Germany) or Hygo-C-Cage® (Privelop Spine, Ennetbürgen, Switzerland)) or titanium mesh cage in cases of corpectomy was subsequently placed. The self-retaining cervical retractor was readjusted to the next segment. After fitting the segmental cages and/or bone-filled titanium-mesh cage, a multilevel semi-constrained lordotic anterior cervical plate was fixed with segmental screws. The screws were placed under fluoroscopy. In order to provide good biomechanical stability, the two superior screws were (and should be) placed just above the interspace, six degrees medially and 12 degrees rostrally, and the inferior screws were placed 12 degrees caudally and six degrees medially. The screws were fastened to the plate with the appropriate locking mechanisms. The indications for additional posterior cervical fixation included three- and four-vertebrae corpectomy, complex fractures, and anticipated non-union due to low bone quality or cervical osteomyelitis.

### 2.3. Statistical Analysis

Data sets were analyzed using the statistical package SPSS version 23 (SPSS, Inc., Chicago, IL, USA). Categorical variables were compared using Fisher’s exact two-tailed test and Pearson’s χ^2^ test, while continuous variables were compared between groups using the Mann–Whitney *U* test. *p* < 0.05 was considered to be significant.

## 3. Results

### Operative and Postoperative Results

Seventy-three patients had four, 10 patients had five, and two patients had six anterior cervical-level fixations. The mean operating time was 3.3 (1.2–6.2) hours. The mean length of hospital stay was 12.8 (4–50) days. Additional primary planned posterior fixation was performed in 10 patients due to anticipated non-union. Secondary posterior fixation was necessary in three cases due to pseudarthrosis. Anterior cervical plate failure with screw loosening and subsequent caudal plate migration following a three-vertebrae corpectomy was observed in two women aged 48 and 63 years. Both cases occurred postoperatively within 4 months. Both patients underwent a C4, C5, and C6 corpectomy for the treatment of multilevel degenerative cervical myelopathy. Anterior revision surgery with plate removal and supplementary posterior fusion was necessary in both cases. The third patient had additional posterior fixation because of non-union after a hybrid fixation involving a two-vertebrae corpectomy and one segmental cage fixation without anterior plating. In total, 8% of patients who had four-level anterior cervical fixation surgeries underwent supplementary posterior fixation. In the five- and six-level groups, posterior fixation was performed in 50% and 100% of the cases, respectively. During long-term follow-up including 67 patients with follow-up times longer than two years and 7 patients longer than 1 year, there were no further cases of pseudarthrosis.

There were no severe intraoperative complications such as vertebral artery injury or dissection. An early postoperative epidural hematoma after a five-level anterior cervical decompression and fusion (ACDF) of C2 to C7 occurred in a 61-year-old woman resulting in acute tetraparesis due to spinal cord compression. Following a rapid revision decompression surgery, there were no lasting neurological deficits. Two days after a hybrid construct comprising of a two-vertebrae corpectomy and a one-level intervertebral cage fusion and subsequent anterior plating C3 to C7, the 59-year-old male patient under aspirin medication developed soft tissue swelling and severe difficulties swallowing. The symptoms resolved after successful evacuation of the hematoma. A 68-year-old patient with diabetes suffered a cervical soft tissue abscess 2 months after a four-level ACDF surgery with plating C3 to 7. Anterior revision surgery was performed and *Streptococcus anginosus* was isolated. There was no evidence for a pharyngeal or esophageal injury. The patient underwent additional prophylactic supplementary posterior fixation and a long-term antibiotic therapy was administered.

Postoperative C5 palsy was evident in seven patients with complete remission in four patients 3 months postoperative. Postoperative dysphagia and hoarseness occurred in 11 patients, nine of which reported complete remission at discharge. Dysphagia persisted in two patients during the long-term follow-up period. Interestingly, there was no noted permanent recurrent nerve damage. Four patients with chronic obstructive pulmonary disease required a transient tracheostomy due to pulmonary complications resulting in delayed weaning, two of which had worsening of the postoperative mJOA scores.

The mean preoperative VAS for neck and arm pain was 6.9, which improved postoperatively at the last follow-up to 1.3 (*p* < 0.001). The mean mJOA score improved from 13.9 to 16.5 (*p* < 0.001) at the last follow-up. At the last follow-up, 61 patients (81%) with preoperative upper limb sensory dysfunction showed significant resolution of symptoms. Twenty-eight patients (65%) showed improvement in upper motor function. Twenty-eight out of 34 patients (82%) with preoperative gait disturbance demonstrate postoperative improvement. In 5 out of 13 patients, sphincter dysfunction completely resolved during the last follow-up. The mean preoperative Cobb angle was 5.7° and postoperative angle was 17.6° (*p* < 0.001). (Table 2)

## 4. Discussion

Our results confirm recent series [6,7,10,11,12,13,14,15,16] in which the anterior approach for the treatment of more than four levels (cervical) is safe and effective in the treatment of multilevel cervical degenerative myelopathy and radiculopathy, in addition to multilevel cervical spondylodiscitis, complex traumatic cervical fractures, metastatic cervical spine tumor, and ossification of the posterior longitudinal ligament.

In cervical degenerative myelopathy and radiculopathy CT and MRI demonstrate ventral compression of the spinal cord in >75% of patients [15]. Ossification of the posterior longitudinal ligament (OPLL) is a process of fibrosis and calcification, and spinal OPLL may involve the spinal dura. OPLL occurs behind the vertebral bodies and disc spaces as a continuous lesion. Compression of the anterior spinal cord may be focal or extensive, involving the entire cervical spine [17]. Subaxial cervical deformities occur most commonly in the sagittal plane, primarily as a kyphotic deformity. Kyphosis may develop secondary to advanced degenerative disease, trauma, and/or metastatic neoplasia. Failed surgery kyphotic deformities are very common [3,18]. Postoperative development of cervical kyphosis may follow both ventral and dorsal cervical approaches. After anterior cervical decompression, kyphosis may develop secondary to pseudarthrosis or failure to restore anatomical cervical lordosis during surgery. After laminectomy without fixation, kyphosis may develop and progress secondary to disruption of the natural stabilizing structures. The incidence of kyphosis in this situation has been estimated to be as high as 21% [3,4,5,18,19].

The choice of surgical procedure for cervical spine treatment is dictated by several factors, including pathology (degeneration, OPLL, infection, tumor), pathology location (ventral, dorsal, or circumferential), pathology extent (limited to interspace or extensive behind the vertebral body), patient’s medical condition, presence of instability and kyphotic deformities, and familiarity of the surgeon with the procedural techniques. The procedure should be chosen to decompress the affected spinal cord or nerve roots, maintain or restore spinal stability, and correct or prevent kyphotic deformities. The procedure must be tailored to the individual patient’s pathology and medical condition in addition to the surgeon’s comfort with a given procedure [9,18,20,21]. There are several advantages of the anterior approach for the cervical spine. It allows anterior decompression of the spinal cord and nerve roots, maintenance of disc height, correction of hypermobility, and restoration of cervical lordosis [4]. For cervical degenerative myelopathy caused by compression at multiple levels, it is important to assess sagittal balance and location of the offending pathology. If the compression is caused by large disc herniations or osteophytic bars, a dorsal decompression will still result in spinal cord compression, especially in the setting of a straightened or kyphotic spine. Cervical kyphosis is difficult to correct from a dorsal approach [4,5,8,18]. Kyphosis correction requires increasing the anterior column height and decreasing the posterior column height. Multiple level discectomy and fixation has the advantage of being capable of increasing the anterior column height by cervical disc and/or vertebral body replacement [4,5,8,18]. In our series, we had significant lordosis restoration. The symptoms related to both spinal cord and nerve root compression, such as myelopathy assessed by the mJOA score and postoperative pain assessed by the VAS, improved. The mJOA score is a validated, diagnostic tool that is increasingly being used to measure baseline myelopathy severity, postoperative improvements, and social independence [22]. In our series, we had an improvement in the mJOA score from 13.9 to 16.5. This is in accordance to other main series [15,16,23].

However, some biomechanical principles are essential to avoid complications, including pseudarthrosis or construct failure. The use of anterior cervical plating in multisegmental anterior fixation surgeries is essential for improving the fixation rates, especially in multilevel constructs [3,24,25]. In 83 out of the total 85 cases in this series, anterior cervical plating with semi-constrained locking plates was used. One of the two exceptions developed pseudarthrosis, and a posterior fixation was subsequently performed after which union was achieved.

It is true that fusion rates with a one- and two-level corpectomy are higher than two-level or three-level discectomy, although corpectomy has a higher level of graft displacement or extrusion [4,26]. Therefore, in four and more levels, multiple discectomies should be preferred instead of multiple sequential corpectomies. At three or greater levels of corpectomy with ventral instrumentation, failure rates rise significantly up to 70% [26]. Graft displacement after corpectomy is proportional to graft length and is increased with fusion ending at the C7 vertebral body. Although there are many more interfaces between the vertebral bodies and the interbody grafts, multiple level discectomy and fusion provides more points of fixation to hold the construct rigidly in place for longer constructs [13,27,28]. With a corpectomy and titanium mesh cage, there are only two interfaces between the vertebral bodies and the graft occurring at the remaining rostral and caudal vertebral bodies. There is more translational movement at these interfaces because the construct only allows for two points of fixation. This may be the mechanism for the increased complication rates and lower fusion rates seen in longer corpectomy constructs. These findings were confirmed in a recent meta-analysis [29]. In our series, corpectomy was only performed in cases of vertebral body destruction by either infection or tumor. Additionally, this procedure was reserved for those patients with anterior spinal cord compression in whom disc space and osteophyte excision above and below the vertebrae were inadequate to decompress or for patients with an irreducible kyphotic deformity in whom the entire body had to be resected to restore alignment and lordosis. However, we experienced two cases of plate-related complications following two- and three-level corpectomies, thus necessitating anterior revision surgery with subsequent posterior fixation. In accordance with the literature, we suggest adding posterior stabilization with multilevel anterior decompression and fusion including two level or greater corpectomies because of the high failure rates [12,30]. In our series, all patients with three- and four-level corpectomies received posterior stabilization.

Other complications of anterior decompression and fusion included vertebral artery injury or dissection, vocal cord palsy, dysphagia, tracheal and/or esophageal injury, and wound infections. Because candidates for multilevel anterior decompression and fusion frequently have a serious underlying disease, such as spondylodiscitis or metastatic cervical tumor, or have undergone multiple prior procedures, the incidence of these complications is often greater. In our series, there was no case of tracheoesophageal or neurovascular complication. However, we observed one case each of soft tissue hematoma, soft tissue abscess, and epidural hematoma. Dysphagia is a fairly frequent symptom after anterior cervical surgery and can be encountered in up to 50% of cases in the immediate postoperative period [31]. Dysphagia is dependent on the number of levels treated [31]. At 12 months postoperative, in agreement with other series, the rate of moderate to severe dysphagia decreased to 3% [32].

Recurrent laryngeal nerve palsy has been reported in 2–11% [33]. Routine, postoperative laryngoscopy revealed that the true incidence of initial and persisting recurrent nerve palsy after anterior cervical spine surgery was much higher than anticipated. Complete otorhinolaryngological examinations, including laryngoscopy, was performed in only 14 patients with throat pain, postoperative dysphagia, or hoarseness, and recurrent nerve palsy was excluded. Jung et al. [33] reported that the postoperative rate of clinically symptomatic recurrent nerve palsy was 8.3%, and the incidence not associated with hoarseness was 15.9%. Therefore, many patients with recurrent nerve palsy were probably not diagnosed in our series. However, our results indicate that anterior cervical decompression and fusion should be performed with a left-sided approach and with an additional reduction of the endotracheal cuff pressure while the retractors are inserted. 

An infrequent but serious complication was postoperative C5 palsy with 8% recovering to 3% at the long-term follow-up. Based on a recent meta-analysis of 61 series and 11,481 patients, the rate of C5 palsy was 6.3%, which was comparable to our series. Patients with postoperative C5 palsy generally have a good prognosis for functional recovery.

In our series, four patients with preoperative mild chronic obstructive pulmonary disease suffered from respiratory failure in the postoperative course. The bridging process from long-term invasive mechanical ventilation to extubation was 8–50 days. None of these patients had any intraoperative complications such as extensive blood loss, neural or vascular injury, or an underlying spondylodiscitis or malignant disease. At the last follow-up, all patients showed an improvement in the postoperative mJOA scores as compared to the preoperative scores, except two patients who had long-term postoperative invasive mechanical ventilation. In a recent series, the incidence of pneumonia was 0.45% among more than 11,000 patients after anterior cervical decompression and fusion. The risk of respiratory failure and long-term invasive mechanical ventilation was much lower, which is by far lower than in our series. Besides dependent functional status risk factors for the development of pneumonia, greater age, chronic obstructive pulmonary disease, and greater operative duration could be included [34]. Consecutively, based on our own data, we prefer laminectomy with fusion or laminoplasty in elderly patients suffering from significant comorbidities such as chronic obstructive pulmonary disease and cervical degenerative myelopathy due to multilevel spinal cord compression. If cervical lordosis is present in the neutral position and the compressive pathology is primarily dorsal or circumferential dorsal, laminectomy and fixation or laminoplasty is an effective procedure to arrest or improve neurological deficits. 

## 5. Conclusion

In case of multilevel cervical degenerative myelopathy and radiculopathy, as well as metastatic cervical spine tumor and spondylodiscitis, the anterior approach is optimal for treatment of the ventral compressive pathology. Additionally, it allows restoration of cervical spine lordosis. Limitations of multiple level decompression and fixation include inability to address dorsal pathology and increasing rates of pseudoarthrosis, especially in cases of corpectomy and increasing number of treated levels. Early supplementary posterior fixation in cases in which it is indicated, reduces anterior reoperation rates and increases fusion rates. Besides the well-known complications, respiratory failure has to be considered. In patients suffering from significant comorbidities, such as chronic obstructive pulmonary disorder, a dorsal approach may be the better option.

## Figures and Tables

**Figure 1 jcm-07-00469-f001:**
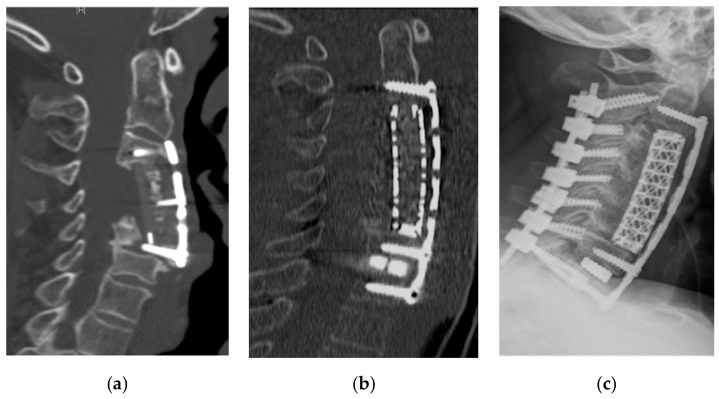
A 60-year-old man’s (**a**) computed tomography (CT) scan showing failed external surgery after two-vertebrae corpectomy and reconstruction with peek cage and anterior cervical plating. (**b**) Postoperative CT after revision surgery with an additional two-vertebrae corpectomy and one segmental cage fixation with anterior plating C2 to T1. (**c**) Postoperative X-ray after early supplementary posterior fixation of C2 to T1.

**Figure 2 jcm-07-00469-f002:**
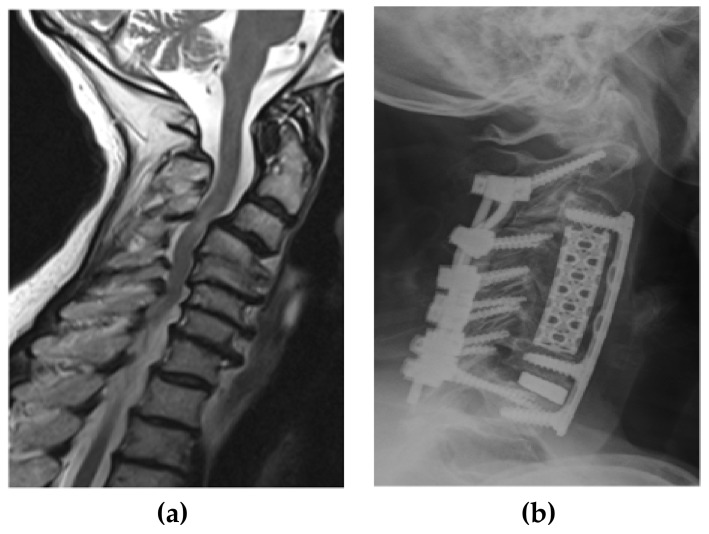
A 74-year-old woman’s (**a**) magnetic resonance image (MRI) showing cervical degenerative myelopathy and degenerative cervical kyphosis, (**b**) postoperative X-ray after anterior cervical decompression and fusion with restoration of cervical lordosis and additional planned posterior fixation due to osteoporosis.

**Figure 3 jcm-07-00469-f003:**
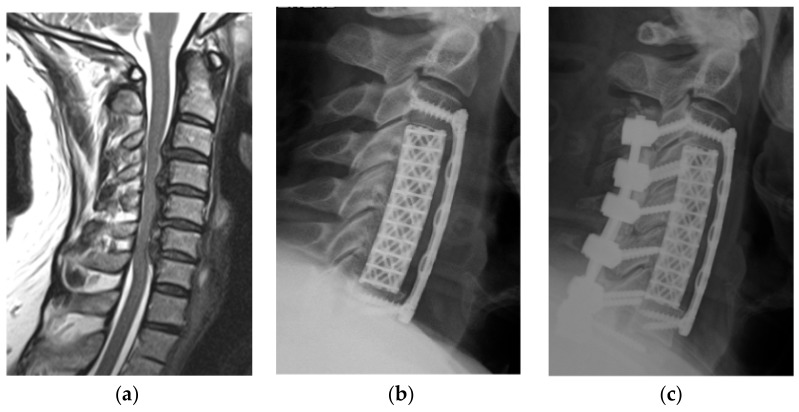
A 49-year-old man’s (**a**) MRI showing ossification of the posterior longitudinal ligament (OPLL) C3 to C7, (**b**) postoperative X-ray after anterior three-vertebrae corpectomy, and (**c**) X-ray after planned posterior fixation.

**Table 1 jcm-07-00469-t001:** Patient characteristics.

Age Mean ± SD (Range) Years	59.6 ± 11.4 (30–78)
Sex F/M	30 ♀ 55♂
Causes	
Cervical degenerative myelopathy and radiculopathy	70 (83%)
OPLL	3 (3%)
Metastatic neoplasia	2 (2%)
Trauma	4 (5%)
Infection	4 (5%)
Implant failure/failed prior surgery	2 (2%)
Comorbidities	
Obesity (BMI > 25)	39 (46%)
Osteoporosis	5 (6%)
Smoking *	13 (15%)
Chronic obstructive lung disease	25 (29%)
Diabetes mellitus	15 (18%)
Coronary artery disease	8 (9%)
Chronic kidney disease	3 (3%)
Levels	
Four-level	73 (86%)
Five-level	10 (12%)
Six-level	2 (2%)
Corpectomy	
One-vertebra	9 (11%)
Two-vertebrae	6 (7%)
Three-vertebrae	4 (5%)
Four-vertebrae	2 (2%)
Duration of surgery (mean ± SD) (range)	
hours	3.3 ± 0.9 (1.5–6.2)
Surgical complications	
Postoperative soft tissue hematoma	1 (1%)
Postoperative epidural hematoma	1 (1%)
Vertebral artery injury	0
Esophageal injury	0
Infection	1 (1%)
Temporary C5 palsy	7 (8%)
Permanent C5 palsy	3 (3%)
Permanent dysphagia	2 (2%)
Anterior construct failure	2 (2%)
Postoperative pulmonary failure	4 (5%)
Dural tear	0
Mortality	0
Follow-up time mean ± SD (range)	
months	16.9 ± 12.1 (3–111)

SD, standard deviation; F, female; M, male; OPLL, ossification of the posterior longitudinal ligament; BMI, body mass index; * currently smokers.

**Table 2 jcm-07-00469-t002:** Pre- and postoperative clinical and radiological parameters.

	Preoperative	Postoperative	*p*-Value
mJOA (median–range)			*p* < 0.001
Mean	13.9	16.5	
SD	2.76	2.29	
Range	(2–17)	(5–18)	
Cervical lordosis			*p* < 0.001
Mean	5.7	17.6	
SD	13.2	9.4	
Range	(−36–32)	(0–40)	
VAS neck and arm (median–range)			*p* < 0.001
Mean	6.9	1.3	
SD	2.47	1.56	
Range	(2–10)	(0–6)	

mJOA, Japanese Orthopedic Association scale; SD, standard deviation.

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
