# Peer review of "Radiological and Clinical Outcome after Multilevel Anterior Cervical Discectomy and/or Corpectomy and Fixation"

_jcm, 2018, doi:10.3390/jcm7120469_

Round 1
Reviewer 1 Report
The authors reported the post-operative outcomes following anterior cervical decompression at ≥ 4 levels. Although the topic discussed is important and relevant, there are a few flaws that compromise the present study. I would raise several examples, 1. I believe most of the surgeons used the term ACDF for “discectomy”. The title and abstract is slightly misleading in that sense. 2. One of the most significant issues in the present study is that the authors discussed multi-level discectomies and corpectomies altogether. As they mentioned in the discussion section, the complication profile of long-segment corpectomy is certainly distinctive and should be discussed separately. For instance, the fusion rate of 1 level corpectomy had been proven better than 2 level discectomy. Given the relative rarity of the complications presented here, I would suggest the authors at least specify which happened in which kind of construct. 3. It appears some of the patients only had very short follow-up period as short as 3 months. When the long-term complications such as pseudarthorosis are to be discussed, minimum 1-year (ideally 2 years) follow-up is mandatory.
Author Response
Dear Reviewer
Thank you for the precise revision and your suggestions which improved the article significantly.
We changed the term decompression and replaced it by discectomy and/or corpectomy.
In our series we wanted to review all relevant aspects associated with multilevel anterior cervical fusion such as underlying diseases and postoperative complications such as hematoma, infections, C5 palsy , pulmonary complications etc. Therefore we included all patients with and without corpectomy in this article.
As suggested we specified our complications in the results part. this is a series of 9 years. During this time span we changed our paradigms based on the experienced complications. All patients with 4 level ACDF receive a plate and all patients with 3 level corpectomy receive additional posterior fixation.
Of course pseudarthrosis rates can only be discussed in cases with follow up longer the 1 year. In our series 67 patients had follow up time of 2 and more years without signs of pseudarthrosis. This is also included in the article.
best regards

Reviewer 2 Report
General
The paper is well-written and provides additional evidence to suggest that multilevel anterior surgery can be performed safely and effectively. Overall, I believe the paper would be a valuable contribution to the literature. Some points for consideration are below:
- Further statistical analysis to determine risk factors for suboptimal outcome would have been a nice addition to the paper. One example for instance - did correction of lordosis relate to mJOA or VAS?
- the authors should consider adopting the degenerative cervical myelopathy terminology for the cases of degenerative disease.
Abstract Background - I do not believe that multilevel anterior cervical decompression is a controversial surgical procedure. There are surgeons who perform such procedures on a regular basis. But it is true that some surgeon prefer to perform posterior surgery when 3 or more levels are involved. (See here for more info — Nouri A, Martin AR, Nater A, Witiw CD et al. Influence of Magnetic Resonance Imaging Features on Surgical Decision-Making in Degenerative Cervical Myelopathy: Results from a Global Survey of AOSpine International Members World Neurosurg. 2017 Sep;105:864-874.)
Methods - How was the mJOA score obtained? Was an mJOA score given retrospectively on patients or was this routinely collected from patients at the time consultation?
Discussion - For “Our results confirm recent series in which the anterior approach for the treatment of >4-levels (cervical) is safe and efficient” I think you mean ‘effective’ instead of ‘efficient’. Also I would reference the AOSpine North America and International studies on degenerative cervical myelopathy, since they have shown an average overall mJOA improvement similar to your study, which helps support your statement of safe and effective. Furthermore, since it is the objective to discuss that multilevel anterior surgery is safe and effective it is suggested that the authors expound a little on their mJOA improvement, for instance what does an improvement of 13.9-16.5 mean? Is this clinically significant (Yes), how does this relate to neurological improvement compared to other studies.
- For “Ossification of the posterior longitudinal ligament (OPLL) is a process of fibrosis, calcification, and spinal OPPL that may involve the spinal dura.” I think you mean ‘OPLL’ not ‘OPPL’, also there is a subtle change needed - the process is ossification rather than calcification; ligamentous calcification is believed to represent a different diagnostic entity. (See here for more info — Takahashi T, Hanakita J, Minami M. Pathophysiology of Calcification and Ossification of the Ligamentum Flavum in the Cervical Spine. Neurosurg Clin N Am. 2018 Jan;29(1):47-54.)
Author Response
Dear Reviewer
Thank you for the precise revision and your suggestions which improved the article significantly.
Regarding the statistical analysis the cohort is to small to determine risk factors as suggested by our statistician. Therefore we included as much information as possible about the complications and the affected patients in the results part.
We performed the Pearson ans the Spearm,ann Correlation and found no correlation between lordosis, VAS and mJOA because of the small cohort. (Lordosis-VAS 0.15; Lordosis-mJOA 0.02; VAS-Lordosis 0.22)
As suggested we changed the terminology (spondylotic to degenerative)
The term "controversial was changed in the abstract.
The mJOA and VAS score were used routinely in all cervical patients since 2008 in our department. We additionally included more information about the patients and their postoperative results. The relevant articles by the AOspine were included in the discussion parts they significantly support our results.
The suggestions regarding OPLL were also included
best regards

Round 2
Reviewer 1 Report
The authors properly amended the points raised in the previous review.